# Targets and Tools: Nucleic Acids for Surface-Enhanced Raman Spectroscopy

**DOI:** 10.3390/bios11070230

**Published:** 2021-07-09

**Authors:** Irene Calderon, Luca Guerrini, Ramon A. Alvarez-Puebla

**Affiliations:** 1Department of Physical and Inorganic Chemistry, Universitat Rovira i Virgili, Carrer de Marcel∙lí Domingo, s/n, 43007 Tarragona, Spain; irene.calderon@urv.cat; 2Institució Catalana de Recerca i Estudis Avançats (ICREA), Passeig Lluís Companys 23, 08010 Barcelona, Spain

**Keywords:** surface-enhanced Raman spectroscopy, DNA, plasmonics, nanoparticles, sensing

## Abstract

Surface-enhanced Raman spectroscopy (SERS) merges nanotechnology with conventional Raman spectroscopy to produce an ultrasensitive and highly specific analytical tool that has been exploited as the optical signal read-out in a variety of advanced applications. In this feature article, we delineate the main features of the intertwined relationship between SERS and nucleic acids (NAs). In particular, we report representative examples of the implementation of SERS in biosensing platforms for NA detection, the integration of DNA as the biorecognition element onto plasmonic materials for SERS analysis of different classes of analytes (from metal ions to microorgniasms) and, finally, the use of structural DNA nanotechnology for the precise engineering of SERS-active nanomaterials.

## 1. Introduction

Surface-enhanced Raman spectroscopy (SERS) is an optical technique that combines nanotechnology with conventional Raman spectroscopy to produce an ultrasensitive, highly specific analytical tool [1,2]. Its integration with spectacular advances in the design and fabrication of advanced nanomaterials [3,4,5] paved the way for the application of SERS in multiple fields such as environmental analysis [6,7,8], materials science [9], data storage [10], forensic science [11], food safety [12], biology [13], and art and heritage [14]. Most notably, in the last decade, SERS biosensing applications have witnessed tremendous growth with a specific focus on clinical diagnosis [15,16,17,18]. SERS-based nanosensors, in particular those integrating multifunctional elements [19,20,21], have the potential to overcome the limitations of conventional methods.

SERS belongs to the class of plasmon-enhanced analytical techniques, as it relies on the excitation of localized surface plasmon resonances (LSPRs) at the surface of nanostructured metals (mostly gold and silver) to enhance the Raman scattering from molecular entities [22]. Excitation of LSPRs with a light of appropriate wavelength causes the localization of highly strong electromagnetic fields at the metallic surface (Figure 1A). Molecules immersed in such intensified local fields experience the first amplification of their Raman scattering (i.e., enhancement of the local field on the analyte, see Figure 1B) whereas the second, multiplicative intensification is ascribed to the successive enhancement of the re-emitted Raman scattering from the analyte [22,23]. As the intensity of these magnified electromagnetic fields rapidly declines with distance from the nanomaterial surface, SERS signals can be obtained, by and large, only from those molecules adsorbed onto or located close to the plasmonic substrate. In other words, SERS is mainly a “first-layer effect”. This mechanism of intensification, which applies regardless of the nature of the molecule, is called electromagnetic (EM) enhancement and represents the *sine qua non* condition to observe SERS. The nature of the plasmon-mediated enhancement is primarily related to the nanostructure features (i.e., composition, size and shape). In this regard, while isolated spherical nanospheres can produce appreciable amplification of the Raman signal, the largest enhancements are observed at the tips of sharp protruding features and, even more so, at nanometric interparticle gaps. The local enhancement at these interparticle “hot spots” results from the plasmonic coupling between nanoparticles and rapidly increases with the shortening of the interparticle gap. Such an effect can be easily visualized in Figure 1A, where the intensity distribution of calculated electromagnetic fields on (i) an individual nanosphere and (ii) its corresponding dimer are compared. SERS amplification of 10^−10^–10^−11^ times can be reached at these gaps [22]. While EM enhancement is the dominant contributor to the SERS effect, additional intensifications of the optical signal can also originate from the change in the Raman polarizability of the analyte upon interaction with the metal surface (i.e., chemical mechanisms). Overall, SERS retains the structural specificity and experimental flexibility of conventional Raman while enabling ultrasensitivity via LSPR-mediated enhancement.

DNA stores genetic information in the living world and this biological role is strictly related to two fundamental structural features of DNA: the complementarity of nucleobase sequences and the double-helical character of the nucleic acid polymer (Figure 1C) [24]. Rapid and ultrasensitive access to genetic information is central in medicine. It is not surprising, then, that nano-optical sensing of nucleic acids for clinical diagnosis has become a major field of research in nanomedicine, intending to overcome the current limitations of conventional analytical methods (e.g., polymerase chain reaction, PCR; and enzyme-linked immunosorbent assay, ELISA) [1,25]. Similarly, the unique recognition properties of nucleic acids (NAs) have been successfully combined with nanomaterials for engineering sensing platforms in a broad range of applications [26,27,28].

In this feature article, we provide an overview of the longstanding, intertwined relationship between SERS spectroscopy and nucleic acids. We begin with the development of SERS sensors for NA sensing through both direct and indirect approaches; then we outline the exploitation of DNA itself as a surface element for SERS-active nanostructures for recognition and quantification of a wide selection of analytes (from metal ions to microorganisms); finally, we highlight the use of the rich chemical functionality of DNA for advanced nanofabrication methods.

## 2. SERS Sensing of Nucleic Acids

SERS sensing strategies can be classified according to two main approaches. In the first, the output signal consists of the intrinsic SERS spectrum of the target analyte (i.e., direct SERS), while the second sensing scheme relies on monitoring the alteration in the Raman signal (e.g., absolute intensity, changes of the spectral profile in terms of relative band intensities, peak frequency and bandwidth) yielded by an extrinsic SERS label (i.e., indirect SERS). It is worth mentioning that terms such as “code”, “probe”, “label”, and “reporter” are generally used as synonyms in the literature. In direct SERS, sensing design is relatively straightforward since it is restricted by the need for direct contact between the target molecule and the plasmonic surface to provide an intense and distinguishable vibrational signal. On the other hand, indirect approaches, where the spectral alterations of the extrinsic SERS labels must be selectively and quantitatively correlated with the biorecognition of the analyte, can be implemented adopting a greater variety of methodologies.

A broad range of diverse direct SERS interrogations of NAs has been reported in the literature [29,30,31,32,33,34]. While intuitively simple, the direct analysis of NAs has posed key challenges regarding the acquisition of intense, reproducible and well-defined SERS spectra. This could be ascribed to different reasons, such as the structural complexity of this class of biomolecules, the diversity of metal surface chemistries that modulate the affinity for NA binding, and the composition of the media [29]. Our group tackled these issues by implementing a new class of positively-charged colloids in place of the traditional negatively-charged ones (e.g., citrate-stabilized silver and gold nanoparticles) [35]. These cationic colloids (AgSp) comprised silver nanoparticles coated with tetramine spermine molecules that imparted positive charge and high affinity for NA as a result of electrostatic binding with negative phosphate groups of NA backbone. Indeed, the addition of tiny amounts of NA to the AgSp colloids promoted the rapid aggregation of the nanoparticles into stable clusters in suspension. Here, the NA sequences acted as molecular bridges between nanoparticles, thereby locating themselves precisely at interparticle gaps where large electromagnetic fields are selectively generated [36]. Figure 2A outlines the DNA-mediated formation of nanoparticle clusters which is also reflected in the illustrative change of the colloidal extinction profile upon NA addition. As a result of such a process, intense SERS spectra with an unprecedented level of reproducibility can be obtained at very low NA concentrations (down to pg levels) [37]. Such capability to gather reliable vibrational fingerprints of the structural properties of NAs, both in terms of composition and conformation, was exploited for the direct SERS detection and quantification of single-strand hybridization into duplex [38] and triplex structures [39], quantifications of the relative nucleobase content [40], recognition of single-nucleotide mismatches and abasic sites [38,41,42], structural discrimination of ribonucleic acids [41], and spectroscopic characterization of the formation of covalent-DNA adducts [28,43]. Most notably, the use of AgSp colloids as plasmonic substrates was found to be particularly suited for the spectroscopic interrogation of duplex structures [44]. As can be seen in Figure 2B, the hybridization of two 21-nucleotide (nt) single-stranded DNAs (ssDNA) into their corresponding duplex (dsDNA) yielded an extended and characteristic set of spectral changes (e.g., peak position, bandwidths, and relative intensity) which put on display their structural reorganization into the double helix via nucleobase stacking and pairing and Watson–Crick hydrogen bonding. Among other phenomena, we highlighted the peak-shifting of the ring breathing modes in the 600–800 cm^−1^ range, which is consistent with base stacking, and of the carbonyl stretching modes (1653 cm^−1^), which is sensitive to hydrogen bonding [38]. Notably, the use of negatively-charged colloids in place of AgSp does not allow the acquisition of SERS spectra that accurately detail the hybridization-induced structural rearrangement of ssDNAs (spectral changes are limited to alterations of relative intensities that solely indicate a variation in nucleobase composition) [44]. The validity of direct SERS analysis of NAs using AgSp colloids in clinical applications has been demonstrated in several studies. For instance, our group devised a SERS-based classification method for differentiating clinically relevant point mutations in 141-nt K-Ras oncogene segments () [45]. Discrimination of such long ssDNAs with single-based sensitivity is afforded by the different secondary structures that these chains adopt under appropriate experimental conditions. In this scenario, AgSp nanoparticles acted as compaction agents promoting the electrostatic-induced folding of the DNA strands into different forms (i.e., A- and B-forms, and a combination thereof). Partial least-squares discriminant analysis (PLS-DA), a well-established and robust classification method, was used to determine the statistically significant difference in the patterns of the SERS spectra. The results demonstrated 100% sensitivity and specificity even for K-Ras sequences with single-base substitution. On the other hand, Trau’s group [46,47] designed an assay for prostate cancer (PCa) risk stratification by combining the AgSp based SERS detection of NAs with an upstream enzyme isothermal amplification (parallel isothermal reverse transcription−recombinase polymerase amplification, RT-RPA). As schematically depicted in Figure 2D, upon extraction of total RNA from urine samples, three RNA biomarkers of PCa (T2:ERG, PCA3, KLK2) were selectively amplified into their corresponding dsDNA forms via RT-RPA which, subsequently, were interrogated by SERS and classified via chemometric analysis with high clinical sensitivity and specificity. It is worth noting that direct adhesion of NAs via their phosphate backbone on the substrate has been also described by Halas and co-workers [48] when using aluminum nanocrystals as plasmonic materials with no requirement of surface modifications.

It is worth noting that the use of more advanced plasmonic substrates and experimental set-ups than the described bulk analysis of aggregated colloids can pave the way for direct SERS analysis of single DNA molecules. For instance, Huang et al. [49] recently demonstrated this possibility by controlling the residence time of gold nanourchins (AuNU), with oligonucleotides adsorbed on their tips, within plasmonic nanopores via an electro-plasmonic trapping effect. The authors demonstrated single-molecule SERS detection of all four nucleobases as well as the discrimination of individual nucleotides in a single oligonucleotide. These results corroborate the potential of SERS to be used as a single-molecule sequencing technique [49,50].

Regardless of the technological advances in direct SERS analysis of NAs, some unavoidable limitations remain intrinsically posed by the direct nature of the method, most notably the need to perform pre-purification steps to isolate the target NAs from complex biological matrices such as real human samples. In fact, in complex media, other molecules can compete with the target sequence for absorption on the plasmonic surfaces, thereby preventing NA adhesion or generating unintelligible SERS spectra resulting from a multitude of different scatterers. Conversely, indirect SERS approaches, despite missing the extensive structural information contained in the intrinsic NA vibrational fingerprints, are inherently more suited for engineering sensing platforms capable of performing SERS analysis directly in the original biological media while simultaneously favoring multiplex and quantitative responses [16,17,51]. Indirect SERS sensing of NAs has typically been carried out using oligonucleotides grafted onto the plasmonic substrates as biorecognition elements (or probes) that selectively bind the target strands. Such an approach has been integrated within a broad range of sensing schemes, exploiting a myriad of diverse SERS substrates. For instance, Barhoumi et al. [52] proposed a hybrid strategy where DNA hybridization was monitored by using oligonucleotide probes equipped with 2-aminopurines (2-APs) in place of adenines. While retaining the same hybridization features, 2-APs yield a very different SERS signal (Figure 3A), thereby enabling label-free detection of the complementary strand by monitoring the appearance of the strong adenine ring breathing band at 736 cm^−1^. However, the most common indirect sensing strategy is one relying on oligonucleotides as “SERS-silent” surface receptors in combination with extrinsic SERS labels to provide signal read-out. A representative example is provided by hairpin-forming structures labelled, at one extremity, with a SERS label (Figure 3B). In this case, the immobilized oligonucleotide forms a stem-loop configuration onto the plasmonic surface, which forces the SERS label to locate close to the nanomaterial (i.e., intense SERS signal: “SERS on”). In the presence of the complementary target strand, the hybridization process breaks the hairpin geometry, thereby separating the SERS label from the plasmonic structure (i.e., “SERS off”). The corresponding decrease in SERS intensity can be then quantitatively correlated with the number of binding events and, in turn, with the concentration of target NAs [53,54]. Another very common sensing design involves the use of SERS-encoded nanoparticles, which typically comprise, as key building units, a plasmonic core as the optical enhancer, surface elements as selective receptors for target molecules, and a SERS reporter/label/code [17,19]. Often, an additional outer layer (e.g., silica) is also integrated into the nanomaterial design to impart high colloidal stability and protect the SERS-labelled plasmonic core from the media, while providing a convenient surface for further chemical functionalization with, among others, oligonucleotides (Figure 3C) [55]. Typically, the presence of the outer shell also prevents an efficient plasmon coupling when the SERS-encoded nanoparticles are brought in close contact, so that the recorded absolute SERS intensity will be proportional to the number of particles in the volume illuminated by the laser. This feature is very convenient for quantitative (multiplexing) analysis. Alternatively, the oligonucleotide-functionalized plasmonic nanoparticles can be devised to actively take advantage of interparticle plasmon coupling to generate larger electromagnetic fields at particle junctions and, in turn, further enhance the SERS signal of molecular reporters positioned at these gaps. As an example, we exploited this strategy to monitor the hybridization of dsDNA and locked nucleic acid (LNA) modified triplex-forming oligonucleotides conjugated onto SERS-encoded nanoparticles comprising spherical silver cores and a resonant SERS label (carboxy-X-rhodamine isothiocyanate; ROX-ITC) [56]. LNA is a structural modification that imparts rigidity and, in turn, increases the binding affinity of the corresponding oligonucleotide for duplexes, enabling the formation of stable triplex structures at room temperature via Hoogsten hydrogen bonds. As outlined in Figure 3D, two sets of silver nanoparticles functionalized with thiolated LNA-modified oligonucleotides were prepared, one of them also labelled with ROX-ITC. Each oligonucleotide was complementary to one half of a 14-bp duplex (dsDNA_00_). The time-dependent triplex-driven nanoparticle assemblies could be monitored via UV-Vis spectroscopy, which revealed a change of the extinction profile due to nanoparticle aggregation. At the same time, interparticle plasmon coupling was also responsible for the increase in SERS signal due to the localization of molecular codes at interparticle gaps. By profiting from the high structural rigidity of triplex structures, duplexes including an internal non-complementary segment of increasing length (dsDNA_05_, dsDNA_10_, and dsDNA_15_; consisting of 5, 10, and 15 base pairs, respectively) were also targeted to promote the formation of nanoparticle clusters with well-defined gap separation (at the nanometric level). This was confirmed by time-dependent bulk analysis of the gap-plasmon resonances in the extinction spectra and overall SERS intensification, which progressively decreased for longer duplexes (i.e., the shorter the interparticle gap, the higher the SERS intensity). The high reproducibility of the time-dependent behaviour of the bulk SERS signal, despite a mixture of clusters of different sizes in the sample, demonstrated effective discrimination between different duplex lengths. On the other hand, the triplex-DNA ability to engineer precise nanoscale gaps was exploited to unravel the relationship between the dynamic structural properties and the bulk plasmonic and SERS responses of randomly aggregated nanoparticles in suspension. Most notably, the results suggest that maximum SERS intensifications are achieved when poorly enhancing individual NPs are assembled into small clusters (ca. 2–6 NPs per cluster) and before larger aggregates are subsequently formed, which occurs at a different time for each duplex. Regardless, bulk SERS enhancements are primarily determined by the interparticle distance, whereas the average aggregation state appears to play a secondary role.

## 3. DNA as Biorecognition Element for SERS Sensing

Besides the conventional use of short thiolated oligonucleotides as surface receptors for targeting complementary NA sequences, we have witnessed in the last decade tremendous growth in the use of aptamers as sensing elements in research, including for SERS-based applications [57,58,59]. Traditionally, aptamers are selected from an extensive pool of random NA sequences via a combinatorial technique known as systematic evolution of ligands by exponential enrichment (SELEX). The identified sequences form versatile tertiary structures with flexible structural conformations that allow the selectively binding, with high affinity, of specific targets via hydrogen bonding, base stacking, and van der Waals and electrostatic forces [57]. Broadly speaking, the integration of aptamers into SERS-based applications takes place via the same sensing strategies as for conventional oligonucleotides employed in NA detection, as previously discussed (e.g., direct vs. indirect, label-free or aptamers labelled with a SERS code, integrated into SERS-encoded nanoparticles, etc.). However, SERS sensing via aptamer receptors can be extended to a vast range of different targets, from metal ions and small molecules to large biomolecules and microorganisms. In particular, aptamers have emerged as an efficient alternative to antibodies because of their improved chemical stability, relative ease of à la carte design and cost-effective production, low immunogenicity, and reduced size. On this subject, our group tested the responses of antibody- vs. aptamer-modified SERS-encoded nanoparticles for the online, rapid and ultrasensitive quantification of *S. aureus* in real human fluids [60]. In a first work [61], we designed a microorganism optical detection system (MODS) that integrates microfluidics and SERS sensing for the multiplex identification of bacterial pathogens in serum and blood. Antibody-modified SERS-encoded nanoparticles were fabricated to guarantee high colloidal stability in biological media while simultaneously minimizing the thickness of the outer functional layer on the spherical particles (Figure 4A). This was to permit an efficient interparticle plasmon coupling (i.e., generation of hot-spots) when NPs selectively accumulated on the bacterial membrane via surface proteins binding. As a result, the overall SERS signal recorded when NP-covered pathogens flow through the volume under laser interrogation was further boosted against the background of unbound nanoparticles, thereby facilitating their real-time, efficient and rapid quantification. Similarly, SERS-encoded nanoparticles were subsequently designed by using a ssDNA aptamer specific to Staphylococcus aureus (*S. aureus*) [62] in place of the *S. aureus* antibody MA1-10708. The number of SERS code molecules (code = mercaptobenzoic acid, MBA) per nanoparticle was maintained at a constant level. To compare the binding efficiency, the two sets of particles were separately combined with a serum solution spiked with around 7 × 10^3^ CFUs per mL of *S. aureus* (the number of viable bacteria in a sample is expressed as colony-forming units, CFUs) and the resulting mixtures were passed through a microfluidic device for optical analysis. Both SERS-encoded particles yielded intense SERS signals when exposed to bacteria, which progressively increased over time up to a plateau reached at t = 800 s (Figure 4A). In the case of aptamer-modified nanoparticles, however, the final SERS intensity was approximately 40% larger thanks to a larger affinity for the bacteria membrane. This can be explained as follows: in contrast to antibodies, aptamers are easily and controllably anchored onto the plasmonic surface without undermining their bioavailability, while their smaller size facilitates the formation of more SERS efficient hot-spots (i.e., shorter interparticle gaps).

Aptamers have also been extensively exploited in the SERS-based sensing of toxic metal ions such as Hg(II) [54]. Mercury ions display a high affinity for thymine residues, promoting the formation of Hg(II) mediated homo base-pairs (T-Hg(II)-T). Our group demonstrated the viability of direct SERS analysis with AgSp colloids for elucidating the spectral changes in the intrinsic vibrational profiles of ssDNA upon Hg(II) binding which, in turn, can be also quantitatively correlated with the metal ion concentration in the sample [43]. However, more conventionally, Hg(II) quantification is carried out via indirect methods using labelled thymine-rich aptamers which, upon target binding, undergo structural reshaping over the plasmonic surface (Figure 3B). An illustrative example is provided by the work of Shi et al. [63]. Here, a silver nanoparticle film was functionalized with aptamers labelled with carboxyfluorescein (FAM) as a resonant SERS code. For high surface coverages, the strands adopted a tilted orientation, thereby placing the SERS code further away from the surface (i.e., weak SERS signal, Figure 4B). The FAM-labelled extremity of the aptamer was brought into closer contact with the plasmonic nanostructure when Hg(II) binding promoted the formation of a hairpin structure (i.e., “SERS on”). An additional SERS code (4-aminothiophenol; 4-ATP) was also directly introduced at the nanostructured surface to be used as an internal standard. Thus, Hg(II) quantification was performed by measuring the SERS intensity ratios between the FAM signal (target dependent) and the 4-ATP signal (target independent). This ratiometric approach has been shown to improve the sensitivity and robustness of the quantitative response over a broader linear dynamic range. Furthermore, the authors generated a dual nucleic acid surface layer by integrating a second biorecognition element consisting of a DNA duplex for the simultaneous detection of Pb(II). The dsDNA was formed by the hybridization of a surface-bound Pb(II)-specific DNAzyme strand labelled with 6-carboxy-X-rhodamine (ROX) and the complementary 17DS strand (Figure 4B). The rigid duplex structure positioned the ROX label further from the plasmonic substrate (i.e., “SERS off”). The addition of Pb(II) caused the DNAzyme strand-activated cleavage of 17DS, which allowed for the approximation of the ROX-labelled extremity of the surface-bound single strand to the silver film thanks to the gained structural flexibility. As shown in the corresponding SERS spectrum (Figure 4B), intense and unique bands of individual labels can be easily distinguished and used for multiplex ratiometric analysis.

It is worth noting that the current availability of selective aptamers for metal ion sensing is restricted to very few species, which hampers a more extensive use of this class of NAs in the field. Conversely, an abundant source of molecular receptors for metal ions can be found among the organic reagents employed in classical qualitative analytical chemistry. For SERS sensing purposes, aromatic reagents are preferably selected from among those that undergo extended alterations of their Raman polarizability upon metal ion interaction. Such alterations are directly reflected in the reshaping of their vibrational profile which, in turn, can be quantitatively related to the number of binding events via ratiometric analysis. Thus, in these indirect approaches, these molecular species act as chemoreceptors integrating, within the same structure, the selective receptor for target ions, the SERS transducer, and the internal standard. However, for SERS applications, the chemoreceptor must be anchored to the plasmonic surface while retaining its chelation capability of coordinating target ions. This significantly limits the number of available organic receptors, since binding to the target species often takes place through the same functional groups that promote their adsorption on metallic nanostructures. Our group recently demonstrated that the structural and functional plasticity of dsDNA for non-covalent interaction with small aromatic molecules via intercalative binding can be used to tackle this issue [64]. Specifically, we selected alizarin red S (ARS) as the organic reagent for the quantitative determination of Al(III) and Fe(III), and a short DNA duplex as the chemoreceptor host (Figure 4C). In the absence of dsDNA, ARS directly bound the metallic surface of AgSp colloids via coordination of the same keto and hydroxyl groups that are involved in metal ion chelation. Thus, it is not surprising to see that the SERS profile of ARS directly bound to positively-charged silver colloids did not show any appreciable alterations in the presence of Fe(III) (Figure 4C). Conversely, when previously combined with dsDNA, ARS retained its chelation capabilities, as demonstrated by the extensive spectral change upon addition of the target ion (Figure 4C). Intriguingly, ARS chelation to Fe(III) and Al(III) yielded well-distinguishable spectral fingerprints due to the distinct electronic redistributions resulting from metal ion coordination. This allowed for the simultaneous detection of both metal ions using one single chemoreceptor.

## 4. DNA as a Structure-Directing Molecule for Nanomaterial Fabrication

Finally, DNA has emerged as an outstanding nanotechnological tool for the rational design of nanomaterials with precision at a nanometric scale. In this context, DNA is used as a structure-directing agent by profiting from the programmability of nucleic acid assembly and DNA’s unique chemical properties, structural plasticity, and tunability [65,66]. Structural DNA nanotechnology has been applied to produce a multitude of diverse, hybrid nano-architectures, including plasmonic materials for SERS applications. An illustrative example of the outstanding ability of DNA to assemble metallic nanoparticles into well-defined molecular-like materials (i.e., plasmonic metamolecules) has been recently reported by Zhou et al. [67]. In their work, the authors exploited a general strategy based on DNA origami to prototype the assembly of core-shell silver-gold nanoparticles (Ag@Au) into metamolecules of high structural complexity and tunable SERS response. Six 10 nm diameter gold nanoparticles were attached to the exterior sides of a hexagon tile template obtained by the programmed self-assembly of DNA-origami honeycomb lattices (Figure 5A). These small AuNPs were used as seeds for in situ Ag deposition to yield Ag@Au core-shell nanoparticles of increasing size. The efficiency of the SERS response, monitored by using 4-mercaptobenzoic acid (4-MBA) as the SERS label, showed an increase with the thickening of the silver shell before neighboring nanoparticles began to fuse (i.e., suppression of interparticle hot-spots), as corroborated by theoretical simulations of the distribution of the electromagnetic field (Figure 5A). Further and more structurally complex SERS metamolecules were constructed by tailoring hexagon DNA monomers (core-satellites, dimers, trimers, 1D chains, etc.), revealing a general correlation between increase in structural complexity and SERS response.

Besides its use as molecular directing agent in the bottom-up self-assembly of nanoparticles, DNA has been also employed as an interfacial-active element for guiding gold/silver ions reduction in overgrowth processes [68,69]. Generation of small interparticle gaps is key to maximizing signal enhancement and, thus, fabricating highly bright SERS encoded structures [70,71]. Salt-induced aggregation of labelled nanoparticles is a straightforward approach to generate very active SERS structures, but the traditional poor control over aggregate sizes, geometries, and gap separations yields highly heterogeneous cluster-to-cluster SERS responses. To tackle these limitations, Nam and coworkers [68,69] proposed an approach based on thiolated DNA-base chemistry for the high yield fabrication of core-shell gold nanoparticles with an interior gap of ca. 1 nm. This class of SERS encoded nanoparticles displayed very high and uniform SERS enhancements with single-particle sensitivity. The fabrication scheme is depicted in Figure 5B. The 20 nm citrate-stabilized gold core was functionalized with thiolated ssDNAs equipped with an internal SERS label (Cy3). The nucleobases acted as a template for the subsequent growth of the gold shell layer and, thus, enabled the precise entrapment of the SERS label at the internal nanogap. Among other variables, the nucleobase composition played a central role in determining the structural property of the core-shell particles, as clearly shown by the use of different thiolated homopolymeric sequences (poly A, poly C, poly G and poly T, see Figure 5B). poly A and poly C-modified gold cores enabled the formation of uniform ~1 nm nanogaps for nearly all particles. On the other hand, small nanohole-like gaps were generated using guanine sequences (nanoparticle aggregation was also detected in this case), while poly T yielded irregularly shaped narrow nanogaps in popcorn-like Au shell structures (Figure 5B). The formation of G-quadruplexes in poly G and the poor binding affinity of thymines for Au have been indicated as the source of such uneven core-shell-like particles. Importantly, upon removal of nucleobases from the sugar moiety and phosphate backbone (Figure 5B, “no base”), no interior gaps were observed within shell growth, further demonstrating the key role of nucleobases in controlled synthesis.

## 5. Conclusions and Future Prospects

In this feature article, we outlined the main facets of the increasingly intertwined relationship between SERS spectroscopy and DNA. SERS offers unique benefits over conventional analytical techniques (e.g., sensitivity, selectivity and multiplexity) while tremendous efforts in nanofabrication and materials sciences have enabled the engineering of a multitude of diverse SERS-active substrates and their integration with other technologies into flexible, multifunctional platforms. As a result, SERS (bio)sensing is continuously expanding its realm of applications into multidisciplinary areas although, despite its exceptional analytical potential, translation into commercial devices is, by and large, yet to be seen. Critical challenges to be addressed are, for instance, (i) large-scale production of robust, reliable and cost-effective SERS platforms, (ii) elaboration of standardized protocols for sample manipulation and measurements, and (iii) spectral analysis. These improvements are essential to fully convert SERS into a truly quantitative analytical technique [1] and, thus, pave the way for converting SERS sensing of nucleic acids, a longstanding implementation of the technique in the academic setting, into viable applications for clinical settings. On the other hand, we also anticipate that foreseeable advances in aptamer technology and product availability will be extremely beneficial for the design of innovative SERS-based sensors for the selective detection of a multitude of different targets, including those that are currently inaccessible or extremely difficult to identify and quantify. Finally, the use of DNA nanotechnology and the programmability of DNA macromolecules for the manufacturing of sophisticated nanostructures with nanoscale precision is an exciting research area for generating customized SERS active materials.

## Figures and Tables

**Figure 1 biosensors-11-00230-f001:**
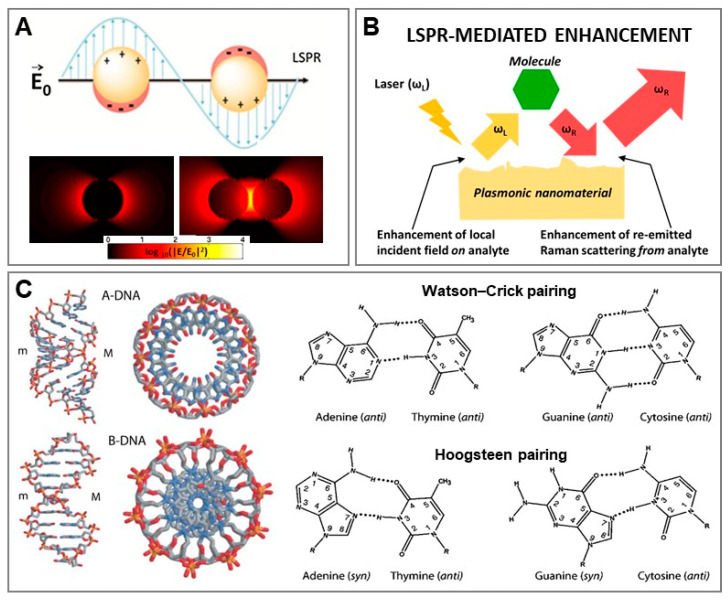
(**A**) Illustration of a localized surface plasmon resonance (LSPR). Calculated electrical fields for a 45 nm diameter silver nanoparticle and its corresponding dimer (interparticle gap, g = 1.31 nm) under a 514 nm excitation laser. Reprinted with permission from [8]. Copyright 2015, American Chemical Society. (**B**) Outline of the plasmonic-mediated enhancement of Raman scattering from a molecule located close to a nanostructured metallic surface (i.e., electromagnetic mechanism). Adapted with permission from [23]. Copyright 2012, Royal Society of Chemistry. (**C**) Structures of A-DNA and B-DNA duplexes (m = minor groove; M = major groove) and A–T and G–C base pairs (Watson-Crick and Hoogsteen pairing). Reproduced with permission from [24]. Copyright 2015, Wiley-VCH.

**Figure 2 biosensors-11-00230-f002:**
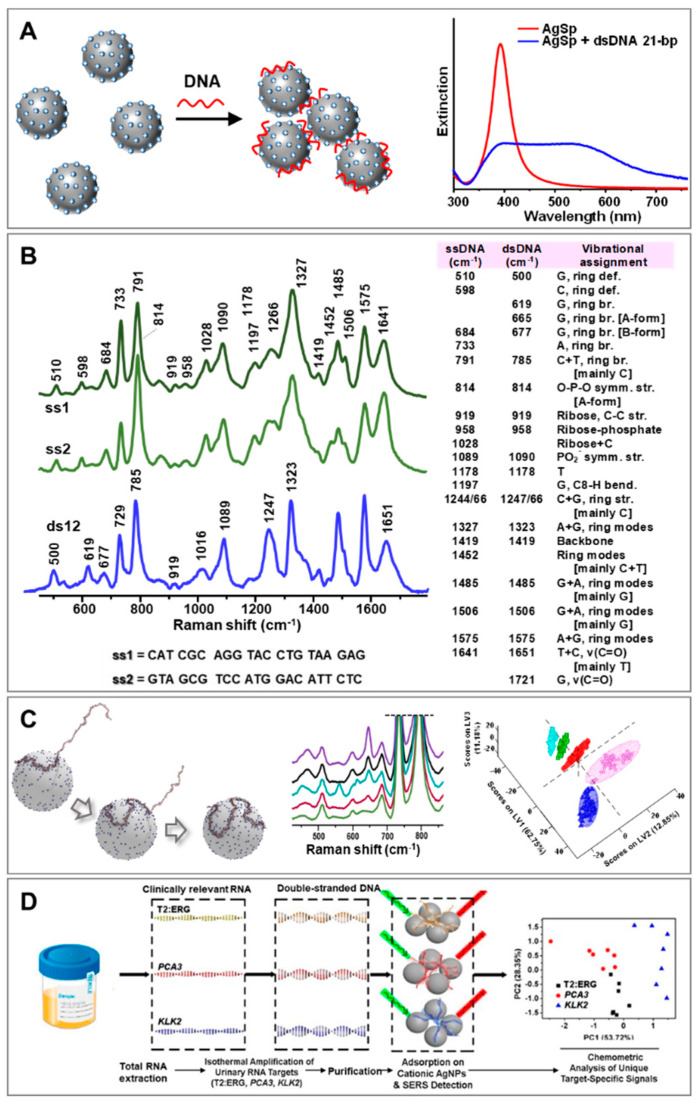
(**A**) Outline of NA-mediated assembly of cationic spermine-coated silver colloids (AgSp) into stable clusters in suspension and the representative extinction spectra of the nanomaterials before and after the addition of a 21-bp duplex. Adapted with permission from [37]. Copyright 2015 WILEY-VCH. (**B**) SERS spectra of two 21-nt complementary ssDNAs (ss1 and ss2) and their corresponding duplex (ds12). A table with the vibrational assignments is also included. Adapted with permission from [29]. Copyright 2018, Royal Society of Chemistry. (**C**) Molecular dynamics simulation of the wrapping process of a 141-nt single-stranded DNA around a AgSp nanoparticle for low salt concentration, SERS spectra of 141-nt ssDNAs with single and double-point mutations in the 430–910 cm^−1^ spectral range, and the resulting partial least-squares discriminant analysis (PLS-DA). Adapted with permission from [45]. Copyright 2017 WILEY-VCH. (**D**) Schematic of SERS detection of urine RNA prostate cancer biomarkers using AgSp colloids and pre-amplification of the target RNAs via reverse transcriptase-recombinase polymerase amplification (RT-RPA). Adapted with permission from [47]. Copyright 2018, American Chemical Society.

**Figure 3 biosensors-11-00230-f003:**
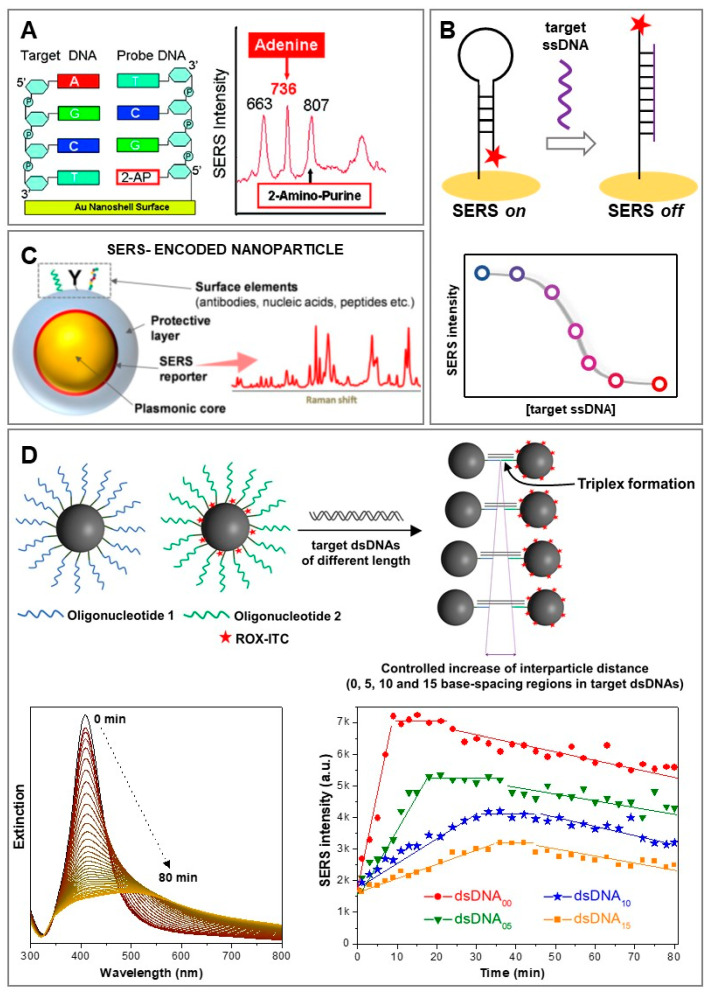
(**A**) Depiction of the label-free sensing scheme for monitoring DNA hybridization: a thiolated oligonucleotide (DNA probe), with 2-aminopurine (2-AP) substituted for adenine, A, is anchored onto a gold nanoshell surface. Upon hybridization with the complementary target DNA, a new adenine band arises in the SERS spectrum. Adapted with permission from [52]. Copyright 2010, American Chemical Society. (**B**) Illustration of a canonical hairpin DNA probe equipped with a SERS label for on-off detection and quantification of the target strand. (**C**) Schematic of a typical SERS-encoded particle. Adapted with permission from [17]. Copyright 2017, Springer. (**D**) Outline of the duplex-mediated assembly of two sets of silver nanoparticles functionalized with triplex-forming oligonucleotides (one of the two batches of nanoparticles was also labelled with carboxy-X-rhodamine isothiocyanate, ROX-ITC, as the SERS code); head-to-head nanoparticle triplex assembly was initiated by the addition of complementary dsDNA of increasing length (dsDNA_00_, dsDNA_05_, dsDNA_10_, and dsDNA_15_). Extinction spectra in the absence (black line) and after the addition of complementary dsDNA between 0–80 min, at 150 s intervals. Time-dependent values of ROX-ITC SERS intensity (peak height of the 1646 cm^−1^ band) for each dsDNA-mediated nanoparticle assembly during the aggregation process. Adapted with permission from [56]. Copyright 2012, Royal Society of Chemistry.

**Figure 4 biosensors-11-00230-f004:**
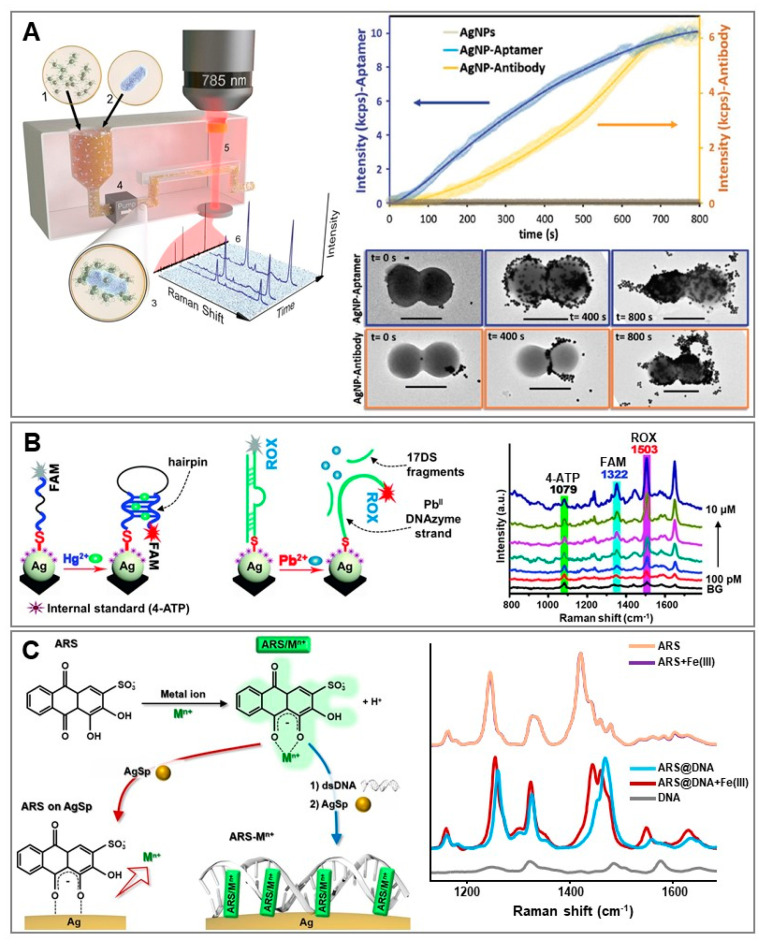
(**A**) Left: Conceptual view of microorganism optical detection system (MODS): SERS-encoded silver nanoparticles labelled with different SERS codes and functionalized with bacteria-selective antibodies are mixed with the biofluid. Selective nanoparticle accumulation occurs on the targeted microorganism membrane. Online SERS interrogation is then performed by flowing the sample into a millifluidic channel. Adapted with permission from [61]. Copyright 2016, Springer Nature. Right: Comparison of the time-dependent SERS intensity of aptamer- vs. antibody-SERS encoded nanoparticles (code = MBA) in serum solutions spiked with ca. 7 × 10^3^ CFUs per mL of *S. aureus* (top). TEM images of *S. aureus* as a function of the incubation time with the SERS-encoded particles (bottom). Scale bar: 1 μm. Adapted with permission from [60]. Copyright 2016, Wiley-VCH. (**B**) Outline of the multiplex SERS detection of Hg(II) and Pb(II) on silver nanoparticle film and SERS spectra of equimolar Pb(II) and Hg(II) mixtures at increasing metal ion concentrations (from 100 pM to 10 μM). Background (BG) stands for distilled water. Adapted with permission from [63]. Copyright 2018, Royal Society of Chemistry. (**C**). Description of the interaction of Alizarin Red S/metal ion complexes (ARS/M^n+^) on AgSp with and without dsDNA. SERS spectra of ARS, ARS + Fe(III), ARS + dsDNA, ARS + dsDNA + Fe(III) and dsDNA in PBS 0.1 M (pH 5.5) in the 1200–1570 cm^–1^ spectral region. Final concentrations in the sample: ARS = 1.75 μM, dsDNA = 48 nM, Fe(III) = 3.5 μM. Adapted with permission from [64]. Copyright 2019, American Chemical Society.

**Figure 5 biosensors-11-00230-f005:**
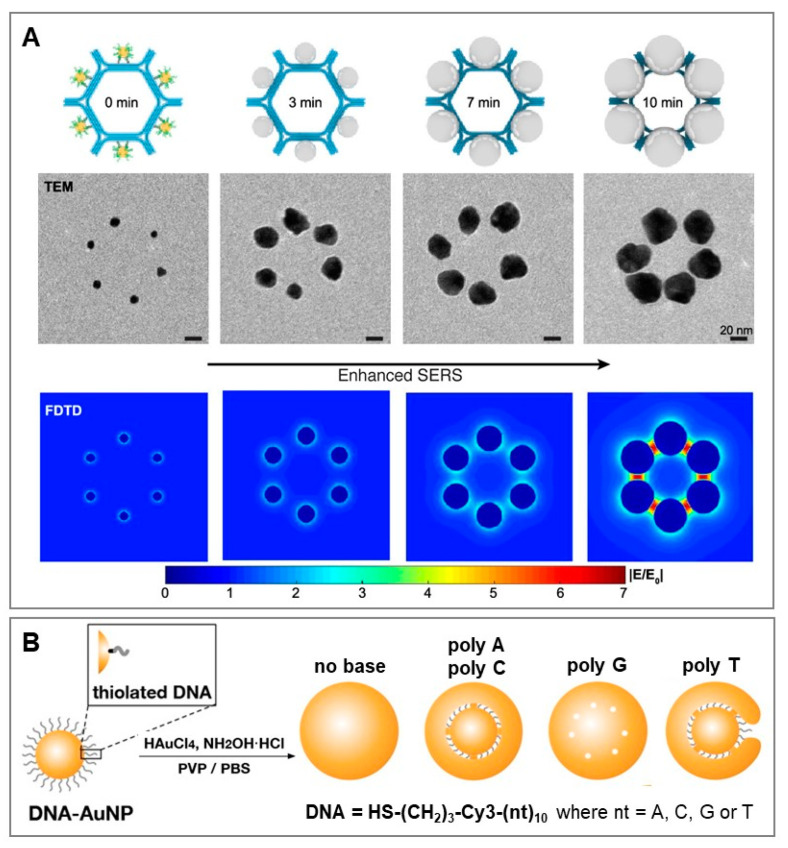
(**A**) Schematic of the fabrication of Ag@Au hexagonal metamolecules on a DNA origami template via control of the growth time of in situ silver deposition (t = 0, 3, 7 and 10 min). At t = 12 min, fusion of the nanoparticle was observed. Representative TEM images and corresponding finite-difference time-domain simulations of the electromagnetic fields (laser = 633 nm) are also included. Adapted with permission from [67]. Copyright 2021, American Chemical Society. (**B**) Outline of the fabrication of core-shell SERS encoded nanoparticle with ultrasmall interior nanogap by exploiting thiolated DNA-based chemistry. The sequences of the thiolated ssDNA poly A, poly C, poly G and poly T are 3′-HS-(CH_2_)_3_-(Cy3)-A_10_-5′, 3′-HS-(CH_2_)_3_-(Cy3)-C_10_-5′, 3′-HS-(CH_2_)_3_-(Cy3)-G_10_-5′, and 3′-HS-(CH_2_)_3_-(Cy3)-T_10_-5′, respectively (where Cy3 = SERS code). “No base” refers to a structurally analogous thiolated ssDNA from which nucleobases were removed while leaving sugar moiety and phosphate backbone intact. Core-shell nanoparticles with nanometric interior gaps originated from DNA-modified gold cores in phosphate buffer saline (PBS) solution with polyvinylpyrrolidone (PVP) as a stabilizer. The reduction of HAuCl_4_ onto the core was performed using hydroxylamine hydrochloride (NH_2_OH∙HCl) as a mild reducing agent. Adapted with permission from [68]. Copyright 2021, American Chemical Society.

## Data Availability

Data Availability upon request to the authors.

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
