# Peer review of "Targets and Tools: Nucleic Acids for Surface-Enhanced Raman Spectroscopy"

_biosensors, 2021, doi:10.3390/bios11070230_

Round 1

Reviewer 1 Report

This review discusses some aspects related to DNA and SERS, in particular the authors report both on the use of SERS to detect DNA molecules (probes, targets, etc.) and on the use of DNA technology to prepare nanostructures for e.m. field confinement to be used as SERS platforms.

The manuscript is well written and to me it sounds interesting.

anyway I think that, considering the main topic, it is missing a very important application of SERS related to DNA. In particular, during the most recent years several groups are trying to use SERS to read DNA molecule at single molecule level and to extract information from that, mainly towards sequencing applications. Moreover, another important aspect discussed in these set of papers regards the reproducible detection of DNA.

I recommend the authors (that also worked on important papers on the topic of ssDNA and dsDNA detection in the past) to include a discussion on these aspect reffering to recent literature (some useful papers are reported below)

Belkin at al. ACS Nano 2015, 9, 10598−10611.

Huang et al. Nat. Commun. 2019, 10, No. 5321.

Hubarevich et al. J. Phys. Chem. C 2020, 124, 41, 22663–22670

Xu et al. J. Am. Chem. Soc. 2015, 137, 5149−5154.

Papadopoulou et al. Chem. - Eur. J. 2012, 18, 5394−5400.

Tian et al. Nano Lett. 2017, 17, 5071−5077.

Reviewer 2 Report

In this paper, the authors overviewed the importance of SERS in studying DNAs as well as the efforts in fabricating DNA nanoparticles for SERS detection. Several representative examples are provided. This is a timely review and is a great contribution to the field. I recommend publication after the following comments have been addressed. 

1. In the introduction part, the authors explained the mechanism of surface enhancement, but offered little information about the technique of Raman spectroscopy. Is spontaneous Raman (with a continuous laser) used or a stimulated Raman technique (with pulsed laser) used in SERS? How to choose the Raman pump? Do we need to consider the resonance enhancement effect? A paragraph to  explain the experimental details of SERS would be very useful.

2. Line 72, should it be Figure 1C?  Line 361, 365, 367, should they be Figure 4C? 

3. The right panel of figure 1A shows a very broad redshifted absorption band, is this due to the formation of nanoparticles? What factors contributes to the broadness of this band? 
